# Duty cycle of 33% increases cardiac output during cardiopulmonary resuscitation

**Taegyun Kim**[1], **Kyung Su Kim**[1]*, **Gil Joon Suh**[1,2], **Woon Yong Kwon**[1,2], **Yoon Sun Jung**[3], **Jung-In Ko**[4,5], **So Mi Shin**[1]

**1** Department of Emergency Medicine, Seoul National University Hospital, Seoul, Republic of Korea,
**2** Department of Emergency Medicine, Seoul National University College of Medicine, Seoul, Republic of Korea, **3** Division of Critical Care Medicine, Seoul National University Hospital, Seoul, Republic of Korea, **4** Department of Emergency Medicine, National Medical Center, Jung-gu, Seoul, Republic of Korea, **5** Department of Emergency Medicine, College of Medicine, Kangwon National University, Chuncheon-si, Gangwon-do, Republic of Korea

* kanesu@gmail.com

**Data Availability Statement:** All relevant data are within the paper and its Supporting Information files.

**Funding:** This work was supported by Seoul National University Hospital (grant number

## Abstract

### Background

The aim of this study was to investigate whether 33% duty cycle increases end-tidal carbon dioxide ($ETCO_2$) level, a surrogate measurement for cardiac output during cardiopulmonary resuscitation (CPR), compared with 50% duty cycle.

### Methods

Six pigs were randomly assigned to the DC33 or DC50 group. After 3 min of induced ventricular fibrillation (VF), CPR was performed for 5 min with 33% duty cycle (DC33 group) or with 50% duty cycle (DC50 group) (phase I). Defibrillation was delivered until return of spontaneous circulation (ROSC) thereafter. After 30 min of stabilization, the animals were re-assigned to the opposite groups. VF was induced again, and CPR was performed (phase II). The primary outcome was $ETCO_2$ during CPR, and the secondary outcomes were coronary perfusion pressure (CPP), systolic arterial pressure (SAP), diastolic arterial pressure (DAP), and right atrial pressure (RAP).

### Results

Mean $ETCO_2$ was higher in the DC33 group compared with the DC50 group (22.5 mmHg vs 21.5 mmHg, P = 0.018). In a linear mixed model, 33% duty cycle increased $ETCO_2$ by 1.0 mmHg compared with 50% duty cycle (P < 0.001). $ETCO_2$ increased over time in the DC33 group [0.6 mmHg/min] while $ETCO_2$ decreased in the DC50 group [-0.6 mmHg/min] (P < 0.001). Duty cycle of 33% increased SAP (6.0 mmHg, P < 0.001), DAP (8.9 mmHg, P < 0.001) RAP (2.6 mmHg, P < 0.001) and CPP (4.7 mmHg, P < 0.001) compared with the duty cycle of 50%.

0420170380), and the grant was given to Gil Joon Suh.

**Competing interests:** The authors have declared that no competing interests exist.

## Conclusion

Duty cycle of 33% increased ETCO$_2$, a surrogate measurement for cardiac output during CPR, compared with duty cycle of 50%. Moreover, ETCO$_2$ increased over time during CPR with 33% duty cycle while ETCO$_2$ decreased with 50% duty cycle.

## Introduction

High-quality cardiopulmonary resuscitation (CPR) is essential for the survival of cardiac arrest victims [1]. Chest compression is a key component of high-quality CPR [2]. Duty cycle is defined as the fraction of chest compression time in an entire chest compression cycle. Although 50% duty cycle is recommended during CPR according to the current guidelines [3, 4], shorter duty cycle that allows increased cardiac filling and cardiac output may be beneficial [5–7]. Duty cycle cannot be modulated during manual chest compressions. However, mechanical compression devices are widely used recently [8, 9], and mechanical compression devices can be modulated to provide chest compression with duty cycle other than 50%.

Pigs are used in various animal researches including sepsis, haemorrhage and cardiac arrest [10–13]. They are especially suitable as experimental animals for cardiac arrest, for they have large chest to accommodate forceful chest compressions and external defibrillations. Moreover, attaching monitoring devices including intravascular catheter is relatively easier in pigs as they are large mammals, and physiologic parameters of pigs and humans are very similar [14].

Our hypothesis was that shorter duty cycle might provide sufficient time for cardiac filling and might result in increased cardiac output during CPR. We performed this study to investigate whether shorter duty cycle is associated with higher end-tidal carbon dioxide (ETCO$_2$) in a swine cardiac arrest model, an index value that positively correlates with cardiac output during CPR [15].

## Materials and methods

### Study setting

This study was a prospective crossover animal experiment. Male domestic pigs weighing 37–43.5 kg were used for the experiment because of their wide use in CPR experiments. All experimental procedures were approved by the Institutional Animal Care and Use Committee of Seoul National University Hospital (IACUC No. 17-0186-S1A2).

### Animal preparation

The experimental animals were purchased for the experiment via the animal laboratory office of Seoul National University Hospital. The experimental animals underwent acclimatisation periods of 14 days before the experiments. The animals were fed with laboratory chow twice a day and were checked by a veterinarian every day. The breeding room temperature was maintained from 18 to 29°C. All the experiments were conducted during daytime at an animal laboratory. Under supine position, the animals were induced for anaesthesia with an intramuscular injection of 5 mg/kg of Zoletil (zolazepam and tiletamin; Virbac Korea, Republic of Korea) and intubated with 6.5 F cuffed endotracheal tubes for mechanical ventilation. Initially, a tidal volume of 10 mL/kg and a frequency of 15/min were applied and adjusted to keep the ETCO$_2$ level within 35–45 mmHg. Anaesthesia was maintained with inhalation of 1%–1.5% of

isoflurane and nitrous oxide. An Arrow Seldinger Arterial Catheter (20 Gauge; Teleflex Inc., USA) and an IntroFlex introducer (8.5 Fr; Edwards Lifesciences Corporation, USA) were inserted into the right common carotid artery and the right external jugular vein with cutdown techniques, respectively. A Swan-Ganz catheter (7.5 Fr; Edwards Lifesciences Corporation, USA) was placed in the right atrium through the introducer. Vecaron Injection (vecuronium; Reyon Pharmaceutical. Co. LTD., Republic of Korea) was continuously infused at a rate of 10 mg/h throughout the experiments. The experiments were performed following haemorrhagic shock and autotransfusion experiments with 145 min-long protocol, and 30 minutes of stabilisation periods were allowed between the two experiments. The animals were under mechanical ventilation throughout the haemorrhagic shock and autotransfusion experiments and the main experiments. We predetermined the humane endpoint for euthanasia as follows: when the systolic blood pressure is below 60 mmHg, or the heart rate is below 50 beats per minute. Animals were planned to be euthanatized with intravenous bolus injection of 40 mEq of potassium chloride if they met the humane endpoint.

## Experimental protocol (Fig 1)

The animals were alternatively assigned to the 33% duty cycle (DC33) group or the 50% duty cycle (DC50) group. A pacing catheter was inserted into the right ventricle, and direct current was delivered with a 9 V battery for 5 s to induce ventricular fibrillation. CPR was started after 3 min of no-flow time. Chest compression was provided using a robot manipulator (VM-6083G model, Denso Co., Ltd., Japan) [16, 17] with a constant rate of 100/min and a constant depth of 5 cm, but with different duty cycles according to the assigned group (phase I). Mechanical ventilation was delivered with a rate of 10/min and a tidal volume of 10 mL/kg during CPR. After 5 min of CPR, 1 mg of epinephrine was injected intravenously, and external defibrillations (biphasic, 200 J) were delivered using a Zoll R Series Defibrillator (Zoll Medical, USA) every min until ROSC. The animals were stabilised for another 30 min after ROSC, they were assigned to the opposite groups, and the same experiments were conducted (phase II).

## Data collection and outcome measures

We collected the following parameters throughout the experiments: body weight, systolic arterial pressure (SAP), diastolic arterial pressure (DAP), right atrial pressure (RAP), heart rate, arterial oxygen saturation and $ETCO_2$. All data were acquired every 2 s using a Vital Recorder program during CPR [18]. The maximal $ETCO_2$ values during every 6 s (i.e. three measurements) were selected because of ventilation rate during CPR. Coronary perfusion pressure

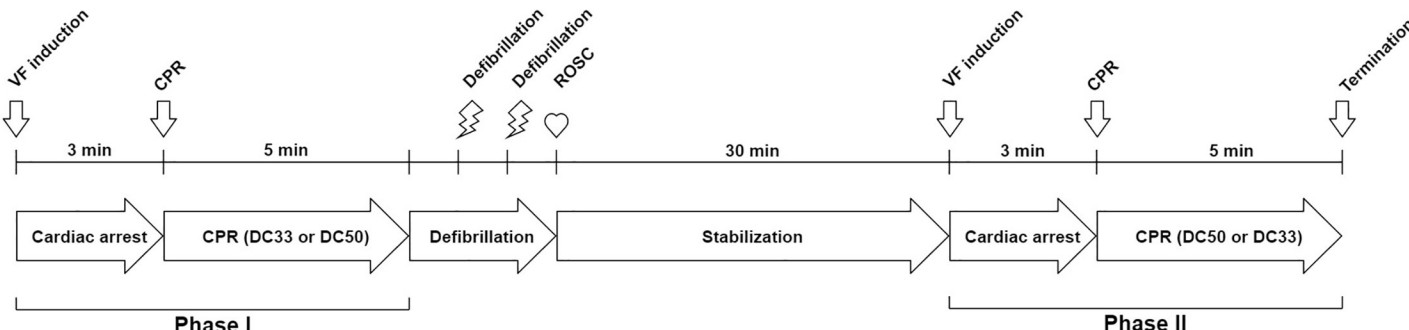

**Fig 1. Study timeline for the animals initially assigned to the DC33 or DC50 group.** Defibrillation was delivered every minute until ROSC was achieved. VF, ventricular fibrillation; CPR, cardiopulmonary resuscitation; ROSC, return of spontaneous circulation; DC33, 33% duty cycle; DC50, 50% duty cycle.

(CPP) was defined as the difference between DAP and RAP. The primary outcome was $ETCO_2$ during CPR, and the secondary outcomes were CPP, SAP, DAP and RAP.

## Statistical analysis

Continuous variables were presented as means ± standard errors or values (95% confidence intervals) and compared using Student's t-tests. We used linear mixed models to test the association among variables including duty cycle, time and outcome variables during CPR. Two-sided P values below 0.05 were considered as statistically significant. All analyses were performed using R version 3.5.1 (R Foundation).

## Results

No animal was euthanatized during the experiments, and all animals finished the whole experimental protocol. Baseline parameters are described in Table 1. Mean $ETCO_2$ during CPR was higher in the DC33 group compared with the DC50 group (22.5 ± 0.3 mmHg vs. 21.5 ± 0.3 mmHg, P = 0.02).

In a linear mixed model, 33% duty cycle affected $ETCO_2$ significantly (P < 0.001), increasing $ETCO_2$ in the DC33 group by 1.0 ± 0.3 mmHg compared with DC50 (Fig 2). The interaction between $ETCO_2$ and time was significant (P < 0.001). $ETCO_2$ increased over time in the DC33 group [0.6 (0.4–0.8) mmHg/min], whereas it decreased in the DC50 group [-0.6 (-0.9–-0.4) mmHg/min].

Duty cycle of 33% increased SAP (6.0 ± 0.7 mmHg, P < 0.001), DAP (8.9 ± 0.3 mmHg, P < 0.001) RAP (2.6 ± 0.3 mmHg, P < 0.001) and CPP (4.7 ± 0.4 mmHg, P < 0.001) compared with the duty cycle of 50%. The interactions were significant between SAP and time [1.9 (1.0–2.7) mmHg/min vs. -0.8 (-1.3–-0.3) mmHg/min, P < 0.001], between DAP and time [0.5 (-0.2–1.2) mmHg/min vs. -0.8 (-1.6–-0.1) mmHg, P < 0.001], between RAP and time [0.2 (-0.4–0.8) mmHg/min vs. 0.7 (0.1–1.2) mmHg, P = 0.02] and between CPP and time [1.0 (0.3–1.6) mmHg/min vs. -1.2 (-1.8–-0.5) mmHg/min, P < 0.001] (Fig 3).

## Discussion

In the present study, 33% duty cycle resulted in higher $ETCO_2$, SAP, DAP, RAP and CPP compared with the 50% duty cycle during CPR. Moreover, 33% duty cycle increased the above values except RAP over time, whereas 50% duty cycle decreased the parameters. Higher $ETCO_2$ value represents better systemic circulation [15] and higher CPP is associated with more ROSC [19], and these results suggest that 33% duty cycle may be more appropriate than 50% duty cycle during CPR.

**Table 1. Baseline haemodynamic parameters at the beginning of the experiments.**

|  | Pig 1 | Pig 2 | Pig 3 | Pig 4 | Pig 5 | Pig 6 |
|---|---|---|---|---|---|---|
| Body weight, kg | 39 | 37 | 39 | 43 | 43.5 | 40 |
| Systolic arterial pressure, mmHg | 95 | 116 | 134 | 107 | 100 | 100 |
| Diastolic arterial pressure, mmHg | 57 | 69 | 39 | 70 | 55 | 62 |
| Heart rate, beat per min | 202 | 187 | 169 | 133 | 147 | 146 |
| Arterial oxygen saturation, % | 100 | 100 | 94 | 98 | 99 | 100 |
| $ETCO_2$, mmHg | 45 | 50 | 48 | 43 | 39 | 36 |
| First assigned group | DC33 | DC50 | DC33 | DC50 | DC33 | DC50 |

$ETCO_2$, end-tidal carbon dioxide; DC33, 33% duty cycle; DC50, 50% duty cycle.

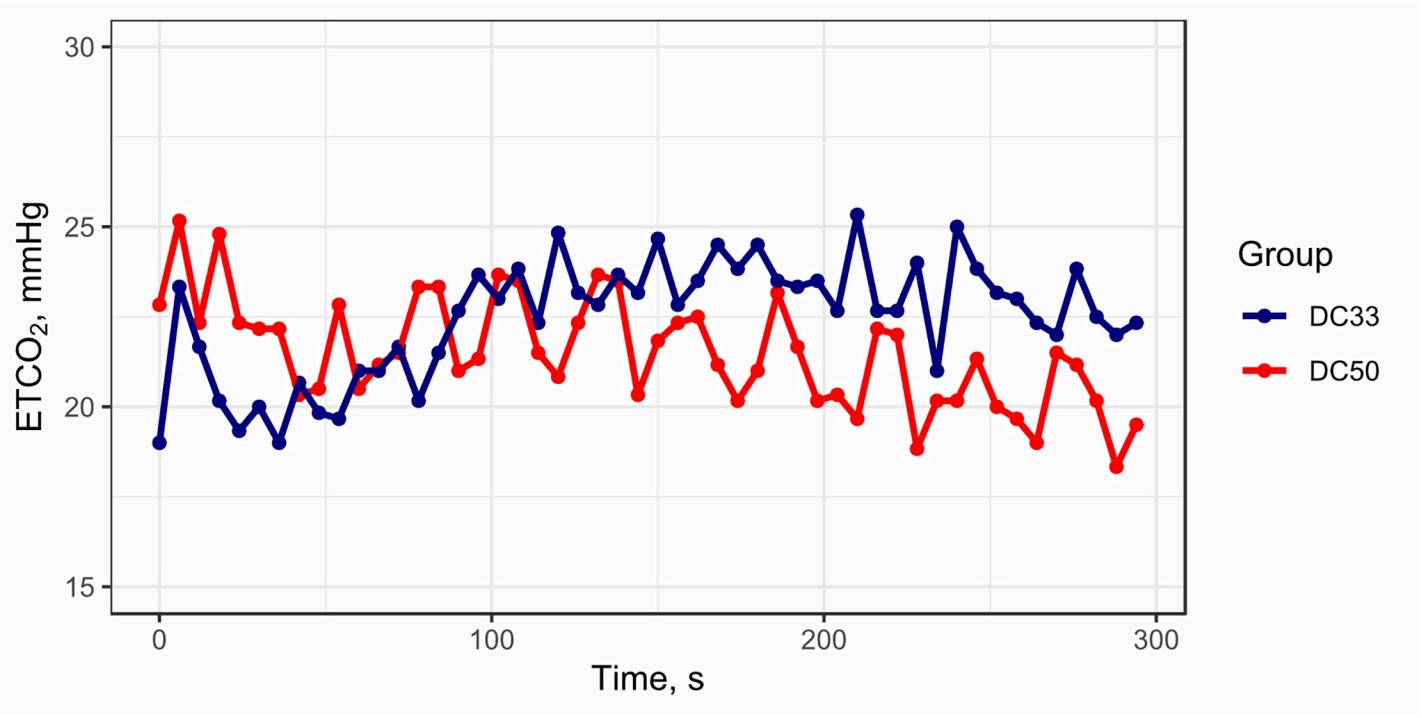

**Fig 2. Mean ETCO$_2$ by group over time.** ETCO$_2$, end-tidal carbon dioxide; DC33, 33% duty cycle; DC50, 50% duty cycle.

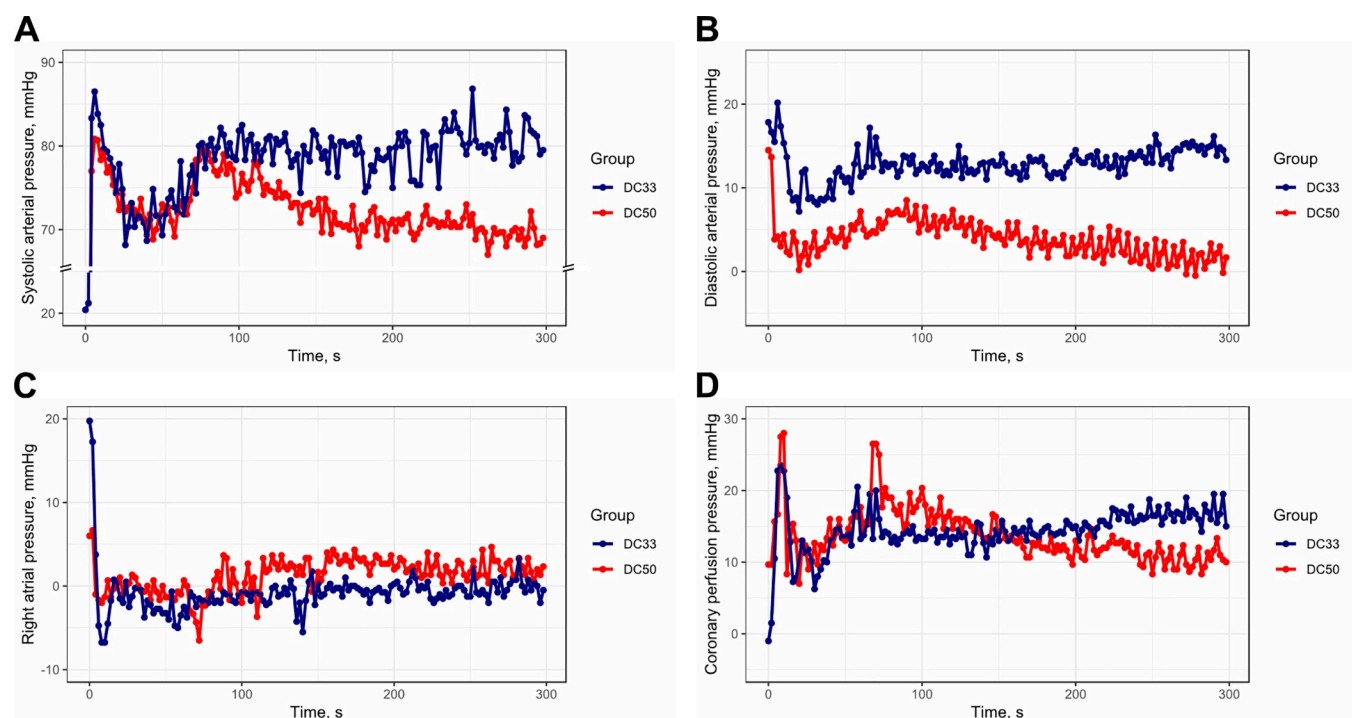

**Fig 3.** (A) mean systolic arterial pressure, (B) mean diastolic arterial pressure, (C) mean right atrial pressure* and (D) mean coronary perfusion pressure* by group over time. *Right atrial pressure and coronary perfusion pressure were missing in two animals of DC33 group (pig 1 and pig 2) and in three animals of DC50 group (pig 1, pig 2 and pig 3). DC33, 33% duty cycle; DC50, 50% duty cycle.

Shorter duty cycle allows enough time for cardiac filling, which leads to higher cardiac output during CPR [5–7]. The results from the present study support the previous studies reporting the association between shorter duty cycle and better haemodynamic parameters during CPR. With widespread use of mechanical CPR and implication of robot CPR in clinical fields in the future, our finding can be considered to modulate the default duty cycle of the above devices.

Although not always consistent, there are several reports on the association between duty cycle and haemodynamic parameters during CPR. Dean et al. [20] have reported that CPR with 30% duty cycle improved myocardial and cerebral perfusion. However, their experimental animals were too young, rather closer to paediatric population, and the compression rates and depths were not constant among the comparison groups. On the contrary, another animal experiment showed no association between reduced duty cycle and coronary perfusion pressure [21], with the limitations that the compression rate was 80/min and decompression depth was 2 cm using an active compression-decompression-CPR device, which might have attenuated the effects of shorter duty cycle during relaxation. Lampe et al. [22] recently reported on the changes in haemodynamic parameters including aortic blood flow, aortic pressure and CPP during CPR according to compression waveforms with different compression rate and different duty cycle, but they have not concluded which waveform is optimal. The compression rate in the study ranged from 50/min to 150/min, and these values are beyond the recommendation from the guidelines [3, 4].

One of the strengths of the current study is that the experiments were conducted according to the current guidelines on adult CPR [3, 4]. We performed this experimental study with fixed compression rate (100/min) and depth (5 cm); therefore, we could also minimize the confounding effects of compression rate, compression depth and compression quality variability according to rescuer fatigue. Finally, we have used the robot manipulator, which provided enough power and precision to carry out the experiments.

Our study has several limitations. First, this was an animal experiment using pigs that once underwent a haemorrhagic shock experiment. The animals might have taken various degrees of ischemic injury during haemorrhagic shock. The design of the preceding experiment was basically a haemorrhagic shock and autotransfusion model, and there is a chance that the pigs were not fully recovered from systemic ischemia-reperfusion injury at the time of the beginning of the duty cycle experiments. We tried to minimize the individual effect from the preceding experiments, and the crossover design might have attenuated the individual differences according to the preceding experiments. Second, we could not measure the rate of ROSC or neurological outcomes because of the design of the study. Moreover, we could not measure cardiac output during the experiments despite insertion of Swan-Ganz catheters. We had to measure RAP continuously, hence we could not locate the tips of the catheters at pulmonary arteries during CPR. We measured $ETCO_2$ and CPP instead, which are the known surrogate markers for cardiac output and ROSC, respectively [15, 19]. Finally, the relatively round shape of the pig chest compared with the humans might have altered the effectiveness of chest compression using the robot manipulator. However, the pigs are used in CPR experiments widely according to their large chests allowing forceful precordial chest compressions [14]. We also tried to maintain consistency in chest compression by tying up the pigs on the experimental table with the ropes and holding the pigs during chest compression. In a previous swine CPR experimental study, the robot manipulator showed stable and effective chest compression even with the alteration in the compression points [17].

We tried to comply with the replacement, refinement and reduction (the 3Rs) principle during the whole experimental process. The pigs were inevitably used for the experiments as our study outcome measures were not able to be investigated with *in vitro* experiments or

simulation studies. We performed the duty cycle experiments in succession to the haemorrhagic shock-autotransfusion experiments, instead terminating the animals to minimise unnecessary sacrifices. The pigs were provided with inhaled anaesthesia to reduce pain.

## Conclusions

In conclusion, duty cycle of 33% increased $ETCO_2$, a surrogate measurement for cardiac output during CPR, compared with the duty cycle of 50%. Moreover, $ETCO_2$ increased over time with 33% duty cycle, whereas $ETCO_2$ decreased with 50% duty cycle.

## Supporting information

**S1 File. The Animal Research: Reporting of In Vivo Experiments (ARRIVE) guidelines checklist.**
(PDF)

**S2 File. The dataset which was used for the analysis.**
(CSV)

## Author Contributions

**Conceptualization:** Taegyun Kim, Kyung Su Kim, Gil Joon Suh, Woon Yong Kwon, Yoon Sun Jung, Jung-In Ko, So Mi Shin.

**Data curation:** Taegyun Kim, Kyung Su Kim.

**Formal analysis:** Taegyun Kim, Kyung Su Kim.

**Funding acquisition:** Gil Joon Suh, Yoon Sun Jung.

**Investigation:** Taegyun Kim, Kyung Su Kim, Gil Joon Suh, Woon Yong Kwon, Yoon Sun Jung, Jung-In Ko, So Mi Shin.

**Methodology:** Taegyun Kim, Kyung Su Kim, Gil Joon Suh, Woon Yong Kwon.

**Project administration:** Kyung Su Kim, Gil Joon Suh.

**Resources:** Kyung Su Kim, Woon Yong Kwon.

**Software:** Taegyun Kim.

**Supervision:** Kyung Su Kim.

**Validation:** Taegyun Kim, Kyung Su Kim.

**Visualization:** Taegyun Kim.

**Writing – original draft:** Taegyun Kim.

**Writing – review & editing:** Taegyun Kim, Kyung Su Kim, Gil Joon Suh, Woon Yong Kwon, Yoon Sun Jung, Jung-In Ko, So Mi Shin.

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
