## [Decision Letter · Decision Letter 0]

9 Dec 2019

PONE-D-19-25910

Shorter duty cycle increases end-tidal carbon dioxide level during cardiopulmonary resuscitation

PLOS ONE

Dear Prof Kim,

Thank you for submitting your manuscript to PLOS ONE. After careful consideration, we feel that it has merit but does not fully meet PLOS ONE’s publication criteria as it currently stands. Therefore, we invite you to submit a revised version of the manuscript that addresses the points raised during the review process.

ACADEMIC EDITOR: All issues raised by reviewers are required.

We would appreciate receiving your revised manuscript by Jan 23 2020 11:59PM. To enhance the reproducibility of your results, we recommend that if applicable you deposit your laboratory protocols in protocols.io, where a protocol can be assigned its own identifier (DOI) such that it can be cited independently in the future. For instructions see: http://journals.plos.org/plosone/s/submission-guidelines#loc-laboratory-protocols

We look forward to receiving your revised manuscript.

Kind regards,

Vincenzo Lionetti, M.D., PhD

Academic Editor

PLOS ONE

Journal Requirements:

3. Please include further information regarding your in vivo study, per our guidelines (http://journals.plos.org/plosone/s/submission-guidelines#loc-animal-research). Specifically, please provide details regarding:

- Source of the pigs

- whether humane endpoints were in place during the study and how they were applied

- the method of euthanasia for the pigs

- any mortality that occurred outside of planned euthanasia or humane endpoints

In addition please complete and submit a copy of the ARRIVE Guidelines checklist, a document that aims to improve experimental reporting and reproducibility of animal studies for purposes of post-publication data analysis and reproducibility: https://www.nc3rs.org.uk/arrive-guidelines. Please include your completed checklist as a Supporting Information file. Note that if your paper is accepted for publication, this checklist will be published as part of your article

We thank you for your attention to these requests.

Reviewers' comments:

Reviewer's Responses to Questions

**Comments to the Author**

1. Is the manuscript technically sound, and do the data support the conclusions?

Reviewer #1: Yes

2. Has the statistical analysis been performed appropriately and rigorously? 

Reviewer #1: Yes

3. Have the authors made all data underlying the findings in their manuscript fully available?

Reviewer #1: Yes

4. Is the manuscript presented in an intelligible fashion and written in standard English?

Reviewer #1: Yes

5. Review Comments to the Author

Reviewer #1: Dear Authors,

it is an interesting study well planned and presented. Here below some observations/suggestions

Abstract conclusion:

I would suggest to add the concept that the 33% duty cycle increased CO. Making soon clear the relation between ETCO2 and CO and/or the clinical relevance of 33% duty cycle could attract more readers.

Title:

considering the above comments, to evaluate also if the title should be modified outlining the effects of 33% duty cycle on CO.

Animal preparation:

were the patients prepared and studied in supine position ?

the fact that the experiment was performed after an haemorrhagic shock could bring to some bias, only slightly touched in the discussion. 30 min of stabilization is not so long and, probably, this aspect merits further discussion.

Experimental protocol:

the mechanical ventilation delivered during the experiment how much time before was started ? how many minutes of stabilization were given ?

data collection:

having a swan ganz inserted, have you collected CO measurements ?

Discussion:

regarding the use of the robot manipulator, has the shape of the pig rib cage given any problem ?

6. PLOS authors have the option to publish the peer review history of their article (what does this mean?). If published, this will include your full peer review and any attached files.

Reviewer #1: Yes: Prof. Edoardo De Robertis, University of Perugia, Italy

---

## [Author Response · Author response to Decision Letter 0]

23 Dec 2019

Thank you for careful review of our manuscript. We reply to your individual comments as follows:

Reviewer #1: Dear Authors,

it is an interesting study well planned and presented. Here below some observations/suggestions

Abstract conclusion:

I would suggest to add the concept that the 33% duty cycle increased CO. Making soon clear the relation between ETCO2 and CO and/or the clinical relevance of 33% duty cycle could attract more readers.

Title:

considering the above comments, to evaluate also if the title should be modified outlining the effects of 33% duty cycle on CO.

Answer: We made small changes in the background and the conclusion section of the abstract (lines 26-28: The aim of this study was to investigate whether 33% duty cycle increases end-tidal carbon dioxide (ETCO2) level, a surrogate measurement for cardiac output during cardiopulmonary resuscitation (CPR), compared with 50% duty cycle.; lines 50-52: Duty cycle of 33% increased ETCO2, a surrogate measurement for cardiac output during CPR, compared with duty cycle of 50%. Moreover, ETCO2 increased over time during CPR with 33% duty cycle while ETCO2 decreased with 50% duty cycle.). We also modified the title as "Duty cycle of 33% increases cardiac output during cardiopulmonary resuscitation."

Animal preparation:

were the patients prepared and studied in supine position ?

Answer: Yes, the pigs were under supine position during the whole experimental processes. We described the position of the animals during the experiments (lines 90-91: Under supine position, the animals were induced for anaesthesia with an intramuscular injection of...).

the fact that the experiment was performed after an haemorrhagic shock could bring to some bias, only slightly touched in the discussion. 30 min of stabilization is not so long and, probably, this aspect merits further discussion.

Answer: We appreciate your comment on the limitation of our study. We described bias due to the preceding haemorrhagic shock experiments in detail in the discussion section (lines 213-225: Our study has several limitations. First, this was an animal experiment using pigs that once underwent a haemorrhagic shock experiment. The animals might have taken various degrees of ischemic injury during haemorrhagic shock. The design of the preceding experiment was basically a haemorrhagic shock and autotransfusion model, and there is a chance that the pigs were not fully recovered from systemic ischemia-reperfusion injury at the time of the beginning of the duty cycle experiments. We tried to minimize the individual effect from the preceding experiments, and the crossover design might have attenuated the individual differences according to the preceding experiments. Second, we could not measure the rate of ROSC or neurological outcomes because of the design of the study. Moreover, we could not measure cardiac output during the experiments despite insertion of Swan-Ganz catheters. We had to measure RAP continuously, hence we could not locate the tips of the catheters at pulmonary arteries during CPR. We measured ETCO2 and CPP instead, which are the known surrogate markers for cardiac output and ROSC, respectively.)

Experimental protocol:

the mechanical ventilation delivered during the experiment how much time before was started ? how many minutes of stabilization were given ?

Answer: We apologize for not presenting enough information in the manuscript. The animals were under mechanical ventilation throughout the preceding experiments and the main experiments. We described the elapsed time for the preceding experiments (lines 101-105: The experiments were performed following haemorrhagic shock and autotransfusion experiments with 145 min-long protocol, and 30 minutes of stabilisation periods were allowed between the two experiments. The animals were under mechanical ventilation throughout the haemorrhagic shock and autotransfusion experiments and the main experiments.).

data collection:

having a swan ganz inserted, have you collected CO measurements ?

Answer: Thank you for your comment. Unfortunately, we could not measure cardiac output using Swan-Ganz catheters because we could not alter the location of the catheter tips during CPR. We commented on not measuring cardiac output during the experiments in the manuscript (lines 221-225: Moreover, we could not measure cardiac output during the experiments despite insertion of Swan-Ganz catheters. We had to measure RAP continuously, hence we could not locate the tips of the catheters at pulmonary arteries during CPR. We measured ETCO2 and CPP instead, which are the known surrogate markers for cardiac output and ROSC, respectively.).

Discussion:

regarding the use of the robot manipulator, has the shape of the pig rib cage given any problem ?

Answer: Thank you for your comment. As your concerns, there might be chances that the shapes of the pig rib cage have affected the effectiveness of chest compression using the robot manipulator. We tied the pigs up on the experimental table and hold them during chest compression. We described on the effective chest compression and the shape of the chest of pigs in the discussion section (lines 225-232: Finally, the relatively round shape of the pig chest compared with the humans might have altered the effectiveness of chest compression using the robot manipulator. However, the pigs are used in CPR experiments widely according to their large chests allowing forceful precordial chest compressions [14]. We also tried to maintain consistency in chest compression by tying up the pigs on the experimental table with the ropes and holding the pigs during chest compression. In a previous swine CPR experimental study, the robot manipulator showed stable and effective chest compression even with the alteration in the compression points [17].)

Finally, we found that we had attached a wrong file as the Fig 3. We substituted the file with a new, correct one. We deeply regret having attached a wrong file.

---

## [Decision Letter · Decision Letter 1]

8 Jan 2020

Duty cycle of 33% increases cardiac output during cardiopulmonary resuscitation.

PONE-D-19-25910R1

Dear Dr. Kim,

We are pleased to inform you that your manuscript has been judged scientifically suitable for publication and will be formally accepted for publication once it complies with all outstanding technical requirements.

With kind regards,

Vincenzo Lionetti, M.D., PhD

Academic Editor

PLOS ONE

Additional Editor Comments (optional):

Reviewers' comments:

Reviewer's Responses to Questions

**Comments to the Author**

1. If the authors have adequately addressed your comments raised in a previous round of review and you feel that this manuscript is now acceptable for publication, you may indicate that here to bypass the “Comments to the Author” section, enter your conflict of interest statement in the “Confidential to Editor” section, and submit your "Accept" recommendation.

Reviewer #1: All comments have been addressed

2. Is the manuscript technically sound, and do the data support the conclusions?

Reviewer #1: Yes

3. Has the statistical analysis been performed appropriately and rigorously? 

Reviewer #1: Yes

4. Have the authors made all data underlying the findings in their manuscript fully available?

Reviewer #1: Yes

5. Is the manuscript presented in an intelligible fashion and written in standard English?

Reviewer #1: Yes

6. Review Comments to the Author

Reviewer #1: I thank the authors to have considered and followed all suggestions/observations to modify the manuscript

7. PLOS authors have the option to publish the peer review history of their article (what does this mean?). If published, this will include your full peer review and any attached files.

Reviewer #1: Yes: Prof. Edoardo De Robertis

---

## [Editor Report · Acceptance letter]

13 Jan 2020

PONE-D-19-25910R1 

Duty cycle of 33% increases cardiac output during cardiopulmonary resuscitation.  

Dear Dr. Kim:

I am pleased to inform you that your manuscript has been deemed suitable for publication in PLOS ONE. Congratulations! Your manuscript is now with our production department. 

With kind regards,

on behalf of

Prof. Vincenzo Lionetti 

Academic Editor

PLOS ONE